# Synthetic prodrug design enables biocatalytic activation in mice to elicit tumor growth suppression

Igor Nasibullin [1], Ivan Smirnov [2], Peni Ahmadi[1], Kenward Vong [3], Almira Kurbangalieva[2] &
Katsunori Tanaka [1,2,4,5 ✉]

Considering the intrinsic toxicities of transition metals, their incorporation into drug therapies must operate at minimal amounts while ensuring adequate catalytic activity within complex biological systems. As a way to address this issue, this study investigates the design of synthetic prodrugs that are not only tuned to be harmless, but can be robustly transformed in vivo to reach therapeutically relevant levels. To accomplish this, retrosynthetic prodrug design highlights the potential of naphthylcombretastatin-based prodrugs, which form highly active cytostatic agents via sequential ring-closing metathesis and aromatization. Structural adjustments will also be done to improve aspects related to catalytic reactivity, intrinsic bioactivity, and hydrolytic stability. The developed prodrug therapy is found to possess excellent anticancer activities in cell-based assays. Furthermore, in vivo activation by intravenously administered glycosylated artificial metalloenzymes can also induce significant reduction of implanted tumor growth in mice.

[1] Biofunctional Synthetic Chemistry Laboratory, RIKEN Cluster for Pioneering Research, 2-1 Hirosawa, Wako-shi, Saitama 351-0198, Japan. [2] Biofunctional Chemistry Laboratory, A. Butlerov Institute of Chemistry, Kazan Federal University, 18 Kremlyovskaya street, Kazan 420008, Russia. [3] Department of Chemistry, The Hong Kong University of Science and Technology, Clear Water Bay, Kowloon, Hong Kong, China. [4] GlycoTargeting Research Laboratory, RIKEN Baton Zone Program, 2-1 Hirosawa, Wako-shi, Saitama 351-0198, Japan. [5] Department of Chemical Science and Engineering, School of Materials and Chemical Technology, Tokyo Institute of Technology, 2-12-1 O-okayama, Meguro-ku, Tokyo 152-8552, Japan. ✉email: tanaka.k.dg@m.titech.ac.jp

In current literature, several prodrug types and their activation mechanisms have been identified and developed (Fig. 1a)[1]. Classically, the efficacy and usage of carrier- and bioprecursor-type prodrugs have long been established. In these cases, activation to bioactive molecules relies on internal mechanisms, such as natural enzymes or specific cellular environments (e.g. acidic pH). With the current progress of targeted delivery techniques and bioorthogonal chemistry[2], however, a burst of interest in the past decade has shifted towards the development of uncaging prodrugs[3]. In general, these types of prodrugs are designed so that relevant functional groups (e.g. −NH₂, −OH, or –COOH) of bioactive molecules are masked, which can then be cleaved in the presence of an external trigger.

As an activating mechanism, transition metal complexes have been heavily explored for numerous uncaging prodrug strategies. This has included the use of masking groups such as allyl carbamates (Ru)[4–19], allyl ethers (Ru)[4,20], and propargyl groups (Pd, Au)[5,6,21–31]. In an alternative approach, the concept of synthetic prodrugs has slowly been gaining attention. These can be described as prodrugs that are activated via bond-forming reactions to construct a drug's backbone (i.e. pharmacophore). It can be reasoned that since the pharmacophore does not exist in the prodrug structure, the overall change in bioactivity following activation should be substantially large. Thus, a theoretical advantage of synthetic prodrugs would be their capacity to be used at higher dosages without eliciting adverse effects.

One of the principal aims of this study was to develop a working example of synthetic prodrugs that can be activated in vivo using artificial metalloenzymes. To tackle this challenge, a concept referred to as retrosynthetic prodrug design was employed. Unlike classical retrosynthetic analysis where the starting point is the target molecule, this approach emphasizes to first identify a suitable reaction that can construct chemical moieties commonly found in drugs. Afterwards, a search is conducted for relevant bioactive compounds in literature, which then becomes the target molecules in the design of synthetic prodrugs.

To date, studies that have successfully reported examples of catalytic bond formation under biological conditions include Suzuki–Miyaura cross coupling (Pd)[9,32], Heck coupling (Pd)[13], alkyne hydroarylation (Au)[33], azide–alkyne cycloaddition (Cu)[34–36], (2 + 2 + 2) cycloaddition (Ru)[37], and olefin metathesis (Ru)[38,39]. To minimize variables during in vivo prodrug activation, intramolecular reactions were prioritized. This ultimately led to the selection of ring closing metathesis (RCM) as a starting point. Furthermore, previous research has shown that following RCM of 1,4,7-trien-3-oles, spontaneous aromatization can proceed[40–43]. This is especially significant since most bioactive compounds contain at least one phenyl moiety in their structures[44]. With the goal of developing a synthetic prodrug that can be activated in vivo, retrosynthetic prodrug design helped to identify the use of sequential RCM/aromatization as an attractive mechanism to construct the core backbone of known aromatic drugs (Fig. 1b).

Herein, we report the design of a prodrug that can be activated through olefin metathesis/aromatization. To conduct this study, precursors to various aromatic pharmacophores will be prepared to identify an appropriate drug scaffold. Optimizations will then address aspects that (1) improve cascade reactivity via enhanced aromatization, (2) improve activity with the biocatalyst, (3)

## a  Types of Prodrug

○ **Physiologically activated**

Carrier prodrugs

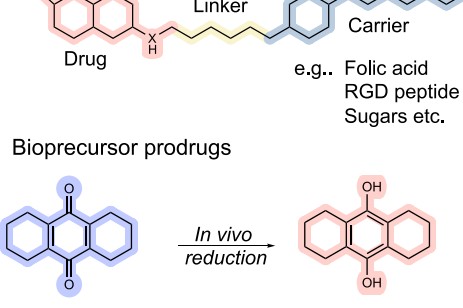

Drug — Linker — Carrier

e.g.. Folic acid
RGD peptide
Sugars etc.

Bioprecursor prodrugs

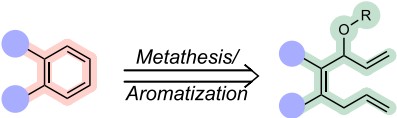

*In vivo reduction*

○ **Externally activated**

Decaging Prodrug

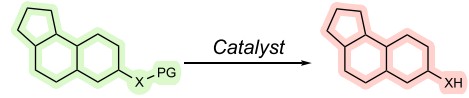

*Catalyst*

Synthetic Prodrug (*This work*)

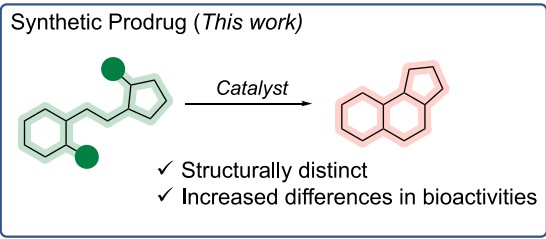

*Catalyst*

✓ Structurally distinct
✓ Increased differences in bioactivities

## b  Retrosynthetic Prodrug Design and Optimization

1) Finding a suitable reaction

*Metathesis/
Aromatization*

2) Structure optimization

○ increase cascade reactivity
(via scaffold choice and leaving group)
○ Increase activity with biocatalyst (via hydrophobic ester)
○ Decrease prodrug effects (via bulky ester)
○ Increase hydrolytic stability (via pivalate ester)

**Fig. 1 Directions in prodrug design. a** General categorization of prodrug types can be made via their activation pathway, which are either naturally- or externally-triggered. This study aims to develop a working example of a synthetic prodrug. **b** Retrosynthetic prodrug design is a concept that aims to design prodrugs that transform into bioactive molecules via a bond-forming reaction. The proposed synthetic prodrug design relies on a sequential RCM/aromatization reaction to construct phenyl-containing bioactive agents.

reduce the intrinsic prodrug bioactivity, and (4) improve hydrolytic stability in blood. Using the optimized prodrug, experiments will then test for anticancer drug activation under in vitro biological conditions (i.e. blood), in cellulo studies using cultured cancer cells, and therapeutic in vivo studies with mice.

## Results

**Pharmacophore backbone screening and optimization.** Although RCM/aromatization is a highly advantageous reaction, one of its perceived limitations is the dependency of slightly acidic pHs (<6) to drive 1,4-elimination to give the final aromatic compound. With this hurdle in mind, the first investigation of this study focused on determining the types of structural scaffolds that can give optimal yields of final product at a physiological pH (~7.4). Shown in Fig. 2, precursors **2–9** that lead to the backbone of various polycyclic and heterocyclic (indole, thiophene, furan) compounds were tested with the simple ruthenium catalyst **1** under aqueous conditions. As expected, RCM activity was generally observed to be high. The $TON_{RCM}$ values, which accounts for both the final product and intermediate, were consistently found within the range of 30–75. However, when specifically looking at aromatization at a pH of 7.4, activity appears to suffer as much lower values for the final products ($TON_{Arom}$) were observed. To better assess aromatization efficiency, the $TON_{Arom}/TON_{RCM}$ ratios were then calculated. From this parameter, an interesting observation was made that showed substrates with lower RCM activities tended to give relatively higher levels of aromatization. For example, substrates **3** and **8** both reach $TON_{Arom}/TON_{RCM}$ ratios of about 1. To explain this, there are likely structural factors that may favor 1,4-elimination but are detrimental to metathesis reactivity. To gain better insight into this phenomenon, further investigations will be conducted in the future.

Focusing on substrates with high RCM activity, one goal of this study was to find ways to further push aromatization yields higher. Generally speaking, a hydroxyl group is a poor leaving group. As such, the importance of the leaving group was next investigated. Using a model naphthalene precursor, three types of leaving groups (**10–12**) were evaluated for their aromatization capacity (Fig. 2). As expected, the most acidic leaving group (ester-containing precursor **12**) showed both excellent RCM activity and full aromatization. In contrast, the ether-containing precursors **10–11** showed insignificant levels of aromatized product. Given this data, it became clear that aromatization at pH 7.4 could be significantly aided by the use of an ester-based leaving group. This observation also consolidates well with results seen in another literature study[40,41].

**Hoveyda-Grubbs-type artificial metalloenzyme (ArM) activity.** Based on the collected data thus far, naphthalene precursors linked with an ester leaving group were chosen as the basis to search for an appropriate prodrug. With this in mind, the next part of the study focused on the substrate compatibility of naphthalene precursors with a biocatalytic system. Artificial metalloenzymes (ArMs) are defined as biocatalysts that incorporate an unnatural metal cofactor into a protein scaffold to elicit an advantageous effect (stereoselectivity, targetability, etc) over the free metal[45–65]. Depicted in Fig. 3a, our group recently developed an albumin-based ArM where Grubbs catalyst **1** was anchored into the hydrophobic binding pocket of human serum albumin (Alb-Ru)[66,67]. Once bound inside the cavity, the metal catalyst was observed to possess remarkable resistance against glutathione quenching. Furthermore, the nature of the binding pocket conferred a degree of specificity for hydrophobic substrates.

To test the ability of Alb-Ru to accept naphthalene precursor **12** as a substrate, reactions were carried out under various conditions (Fig. 3b). In general, the ability of Alb-Ru to facilitate RCM activity was shown to be relatively high, with a TON value of about $31.1 \pm 1.1$ (entry 1). In a similar manner to previous reports[66,68], the presence of 20 mol% of glutathione (entry 2) did not significantly affect ArM activity (TON = $30.8 \pm 0.1$). Experiments were next run to test trends in substrate dependency, where lower substrate concentrations did not seem to adversely affect RCM activity (entry 3, 4). As a further biocompatibility test, reactions were carried out in the presence of cell growth media (D-MEM with 10% FBS). In general, TON values were observed to be lower compared to TONs obtained in buffer solutions. This discrepancy in reactivity is likely a result of the unknown interactions of the biocatalysts and/or substrates with components found in more complex biological solutions. Nevertheless, the preliminary data proved promising enough to carry this study forward.

In the next step of this investigation, improvements to substrate specificity and reactivity were sought using different ester-based precursors. Although the reactivity of precursor **2** seemed initially promising ($TON_{RCM}$ values of 72.0 with free catalyst **1**), its impaired reactivity under lower concentrations resulted in negligible Michaelis–Menten kinetics parameters (Fig. 3d). It can be postulated that the overall hydrophobicity of **2** may not be enough to promote efficient entry into the hydrophobic binding pocket of Alb-Ru. To address this problem, various hydrophobic ester moieties were installed to enhance substrate/enzyme interactions. As a result, precursors **12, 14, 15** were tested to show substantially higher catalytic efficiencies. Notably, the citronellate precursor **14** was found with the highest $k_{cat}/K_M$ (1764 $M^{-1}s^{-1}$). A possible explanation for the greater substrate affinity is likely due to the introduction of a linear $C_{10}$ terpene chain, which mimics fatty acids known to be excellent ligands of albumin.

**Prodrug design and reactivity.** During the literature search for naphthalene-based bioactive agents, combretastatin-A4 (CA-4) and its derivatives were deemed as suitable target molecules for designing synthetic prodrugs[69,70]. Among dozens of these bioactive compounds, **20** was chosen for its structural simplicity and strong bioactivity. In terms of mechanism, all molecules in the family of combretastatins are known to inhibit microtubule polymerization via binding into the colchicine site of β-tubulin (Fig. 4a)[71–73]. Furthermore, in vivo studies have also observed these compounds to disrupt tumor growth via anti-vascular[74] and anti-angiogenic mechanisms[75].

For the next stage of this study, the primary goal shifted to designing naphthylcombretastatin-based prodrugs that can be effectively converted in vivo to the bioactive drug **20** (Fig. 4a). One way to ensure a large difference in bioactivities between the prodrug/drug pair is to ensure the prodrug binds tubulin with low affinity. Since altering the two propene fragments may be detrimental to metathesis activity, derivatization of the ester chain was deemed the most reasonable site. Summarized in Fig. 4a and Supplementary Table 2, modeling studies were first conducted using both (R)- and (S)- enantiomers of the citronellate **16**, pivalate **17**, benzoate **18** and the simple hydroxyl **19** forms of the prodrugs. Compared to the parent drug **20** with an average calculated binding energy of −9.2 kcal/mol, the weakest binder was found to be the citronellate-containing compound **16** (−1.15 kcal/mol). Predicted binding conformations suggest that the steric bulk of the O-linked cleavable moiety can be an important factor in preventing interactions with the colchicine binding site of tubulin.

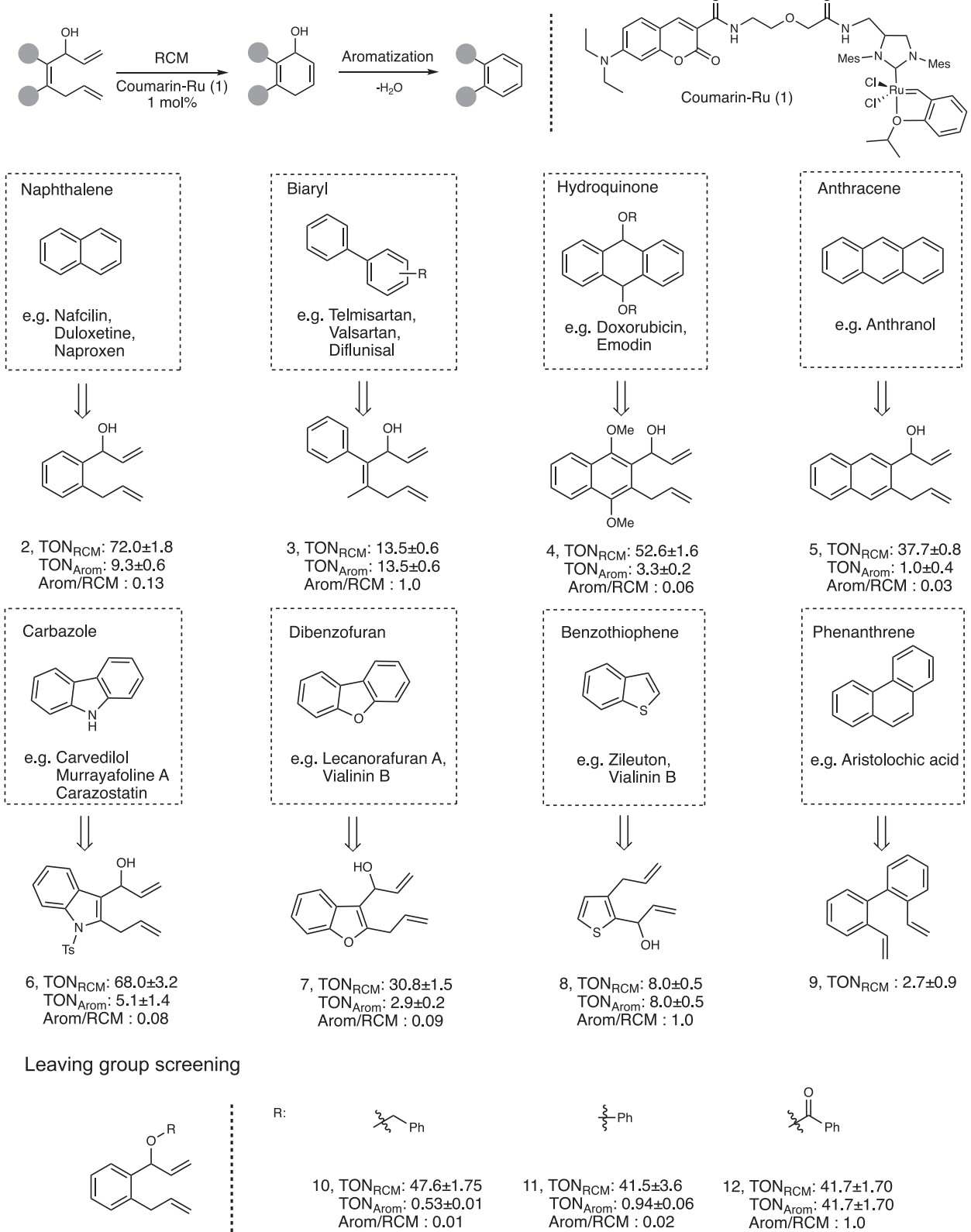

**Fig. 2 Model substrates for various pharmacophore fragments.** All reactions were carried out under the standardized conditions: 4 mM of substrate was incubated with 1 mol% of ruthenium catalyst **1** in PBS/1,4-dioxane (9:1) for 2 h at 37 °C. TON values were determined from product yields obtained by HPLC analysis; where TON$_{Arom}$ refers to final product and TON$_{RCM}$ refers to the combination of cyclohexadien-1-ol intermediate/final product. TON$_{RCM}$ for compound **11** was calculated based on starting material conversion. Reactions were performed in triplicate.

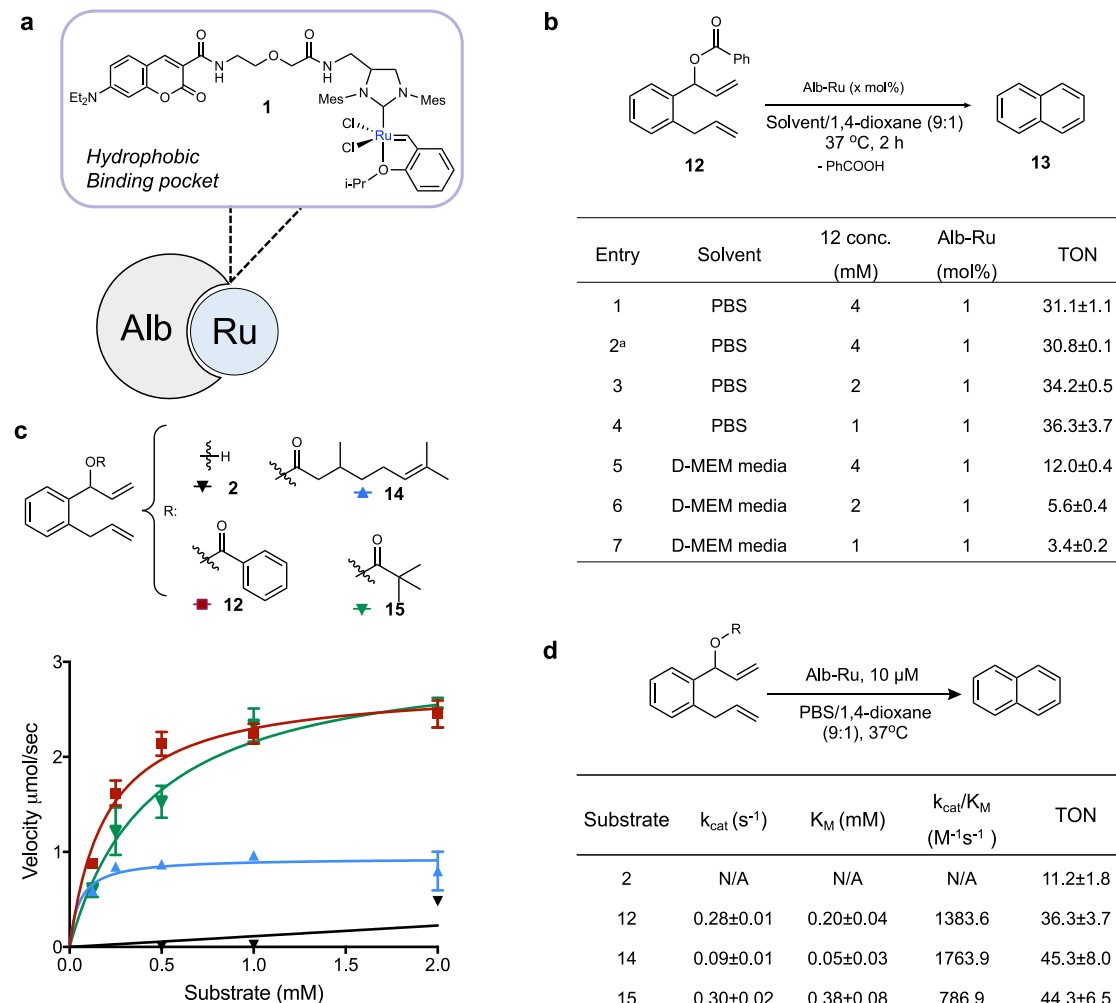

**Fig. 3 ArM-based activity studies with naphthalene-based precursors. a** The anchoring of **1** into the hydrophobic binding pocket of albumin (Alb) leads to the creation of Alb-Ru, which is capable of performing ring-closing metathesis. **b** Reactivity studies focused on the RCM activity of precursor **12** under various substrate concentrations and solvents. $^a$20 mol% of glutathione added. Reactions were performed in triplicate; TON values were determined from product yields obtained by HPLC analysis. Michaelis–Menten kinetic graphs **c** and parameters **d** for substrates (**2, 12, 14, 15**). Summarized values are the substrate affinity ($K_M$), turnover frequency ($k_{cat}$), and catalytic efficiency ($k_{cat}/K_M$). Also presented are the $TON_{Arom}$ using 1 mM substrate concentrations, which were incubated with 1 mol% of Alb-Ru in PBS/1,4-dioxane (9:1) for 2 h at 37 °C. Reactions were performed in triplicate. Data in **c** are represented as mean values ± SD, $n = 3$ independent experiments.

To test prodrug conversion at relevant biological concentrations (<50 μM), a series of reactivity studies at lower concentrations of **16–19** were next carried out. Working with a substrate range of 5 to 250 μM and constant levels of Alb-Ru (10 μM), product yields are shown in Fig. 4b. One of the most striking observations was a perceived activity threshold for the hydroxyl-containing prodrug **19**. Desired product could not be detected at low substrate concentrations (5 and 10 μM), but were observable at levels higher than 50 μM. This is in contrast to the ester-based prodrugs **16–18**, which showed detectable activity even at substrate concentrations of 5 μM. Among them, the pivalate prodrug **17** was observed as one of the most active. As a control, these experiments were also conducted with free ruthenium catalyst **1** (Supplementary Table 5).

Another necessary aspect to consider was the stability of the molecules under biological conditions. Shown in Fig. 4c, hydrolytic stability was evaluated after incubating in blood mixtures at 37 °C for 2 h. When looking at the pivalate **17** prodrug, roughly 9.8 ± 0.7% of the hydrolyzed side product was found. In contrast, the citronellate **16** and benzoate **18** showed 22.3 ± 1.3% and 31.5 ± 1.7% hydrolysis, respectively. And finally, the kinetic activities of the prodrugs were

determined and compared (Fig. 4d). From this data, it can be clearly seen that pivalate prodrug **17** was the better substrate with a higher $k_{cat}/K_M$ (458 $M^{-1}$ $s^{-1}$) compared to the citronellate prodrug **16** ($k_{cat}/K_M$ of 84 $M^{-1}$ $s^{-1}$) and benzoate **18** ($k_{cat}/K_M$ of 108.7 $M^{-1}$ $s^{-1}$).

Taking all the observations into account, it was reasoned that prodrug **17** would be the best candidate moving forward. Its balance between good hydrolytic stability and good catalytic reactivity should give it the best chance of producing biologically relevant drug concentrations in vivo.

**In cellulo efficacy of prodrug therapy**. Moving onto in cellulo studies, control experiments were performed first to evaluate the intrinsic cytostatic activities of the naphthylcombretastatin drug **20** and prodrug **17** against various cancer cell lines (HeLa, A549, PC3, and MCF7). To quantify cytostatic activity, half maximal growth rate inhibition ($GR_{50}$) was determined[76]. Summarized in Fig. 5a, data consistently showed that drug **20** displayed excellent cytostatic activity in the nanomolar range. On the other hand, prodrug **17** is relatively less cytostatic, as its activities are found in the micromolar range. To identify the best model cell line for subsequent anticancer studies, the efficacy of prodrug

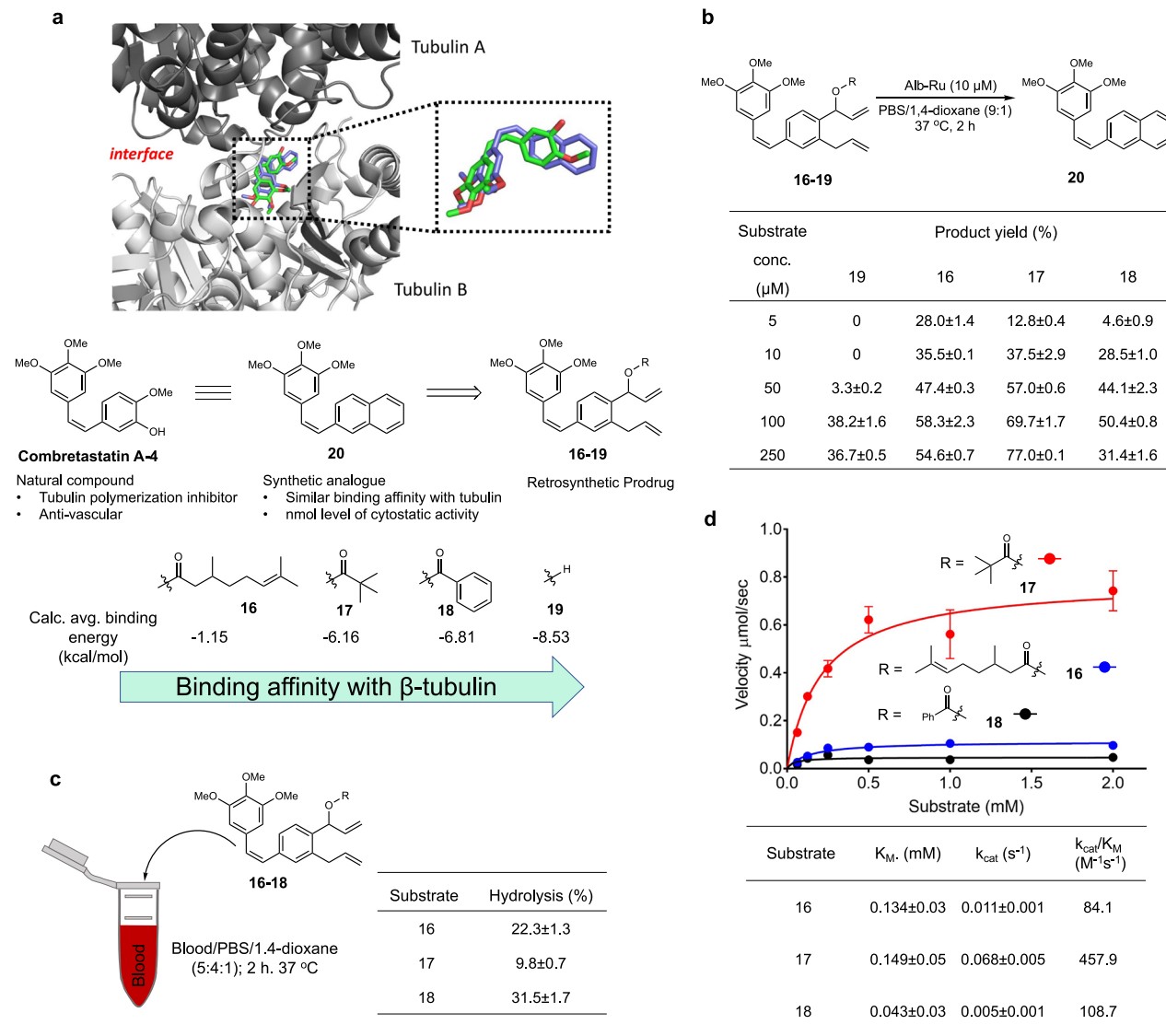

**Fig. 4 Design and development of naphthylcombretastatin-based prodrugs. a** Crystal structure of the dimer interface between tubulin subunits (PDB ID: 5LYJ) showing a bound combretastain A-4 (green), and a modeled conformation of analog **20** (blue). Molecular docking studies of prodrugs **16**–**19** were also conducted to predict their relative binding affinities to the colchicine binding site of β-tubulin (via average calculated binding energies). **b** Alb-Ru activities using biologically relevant concentrations of prodrugs **16**–**19**. Product yield was determined by HPLC analysis. Reactions were performed in triplicate. **c** Incubation of prodrugs **16**–**18** in blood was performed to determine ester stability via the detection of their respective hydrolyzed side product. **d** Michaelis–Menten kinetic graphs and parameters for prodrugs **16**-**18**. Summarized values are the substrate affinity ($K_M$), turnover frequency ($k_{cat}$), and catalytic efficiency ($k_{cat}/K_M$). Reactions were performed in triplicate. Data in (**d**) are represented as mean values ± SD, $n = 3$ independent experiments.

activation was next compared between the cell lines (MCF7 cells were not tested due to relatively weaker drug activity). To perform these experiments, cells were treated with mixtures of Alb-Ru (0.25 or 0.5 μM) and varying concentrations of prodrug **17** (0.125 to 64 μM). From these results, it was determined that the prodrug strategy applied in this study was slightly more effective against HeLa cells ($\Delta GR_{50}$ values are higher than those found with A549 and PC3). Consistent with its known drug sensitivity[70], HeLa cells were thus applied for the following targeted drug studies.

For targeted prodrug activation, a suitable mechanism was needed to direct the biocatalyst to specific cells or tissues within the body. Previously developed by our group, N-glycosylated albumins have been found to be excellent tools for cancer cell targeting[77–83]. Observations showed that HeLa cancer cell targeting could be rapidly achieved using an N-glycosylated albumin decorated with a homogenous assembly of α(2,6)-sialic

acid terminated complex glycans. A possible explanation behind this selectivity likely involves the overexpression of the Siglec-3 (CD33) lectin, which is known to have high affinity towards α(2,6)-sialic acid[84,85]. For the following cancer-targeting studies, a glycosylated artificial metalloenzymes bound with a Grubbs catalyst (GArM-Ru) was used as the mechanism for localized prodrug activation (Fig. 5c). As a note for its preparation, ruthenium catalyst **1** was directed into the hydrophobic pocket of the N-glycosylated albumin, which bound with affinity comparable to the native albumin (Supplementary Fig. 4). To evaluate the in cellulo therapeutic potential of GArM-Ru for the activation of **17** via RCM/aromatization reaction (Fig. 5c), studies were carried out against HeLa cancer cells. To do this, cells were treated with mixtures of prodrug **17** (4 μM), followed by varying concentrations of GArM-Ru (0.03 to 8 μM). Analysis of cell growth then showed that the prodrug therapy to be highly effective under biologically relevant concentrations. This is most evident when

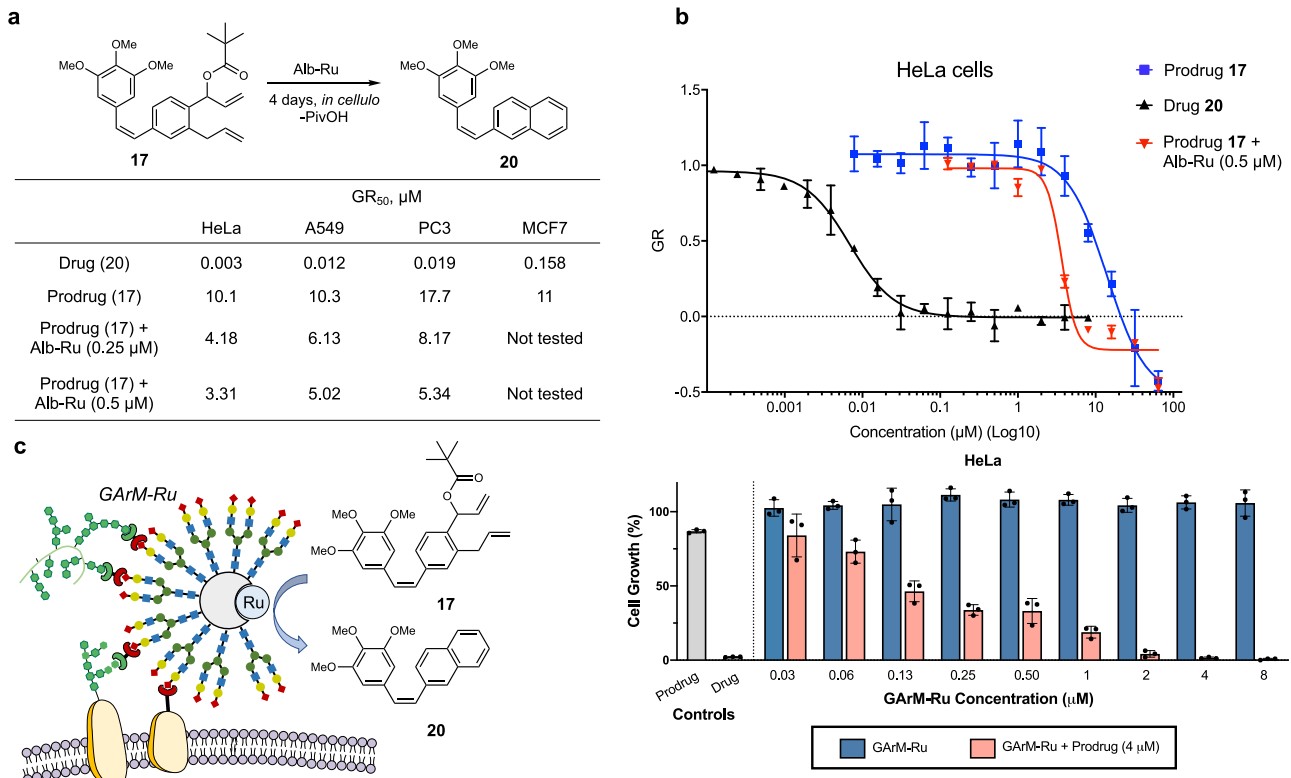

**Fig. 5 In cellulo prodrug therapy against cancer cell growth. a** Summary of calculated $GR_{50}$ values for drug **20**, prodrug **17** and prodrug/Alb-Ru mixtures against HeLa, A549, PC3 and MCF7 cell lines. $GR_{50}$ values represent concentrations that gives half maximal growth rate inhibition. **b** Representative dose response curves against HeLa cells showing the cytostatic activities of drug **20**, prodrug **17**, and mixtures of Alb-Ru (0.5 µM) with prodrug **17**. **c** Schematic depiction of HeLa targeted activation of prodrug **17** into drug **20** using GArM-Ru. The inhibition of HeLa cell growth was investigated using varying mixtures of GArM-Ru and prodrug **17**. In general, the prodrug (and drug) was kept at constant concentrations of 4 µM, while varying concentrations of GArM-Ru was used. Data in (**b**) and (**c**) are represented as mean values ± SD, $n = 3$ biologically independent samples.

only a mixture of 4 µM of **17** and 130 nM of GArM-Ru was needed to cause a ~50% suppression in HeLa cell growth.

**Ring-closing metathesis for in vivo antitumor therapy**. For the final stage of this study, prodrug therapy of the GArM-Ru/**17** mixture was investigated for its capacity to suppress tumor growth in animal studies. Despite our previous report showing that the biocatalytic activity of the GArM complexes could be maintained in the peritoneal cavity of mice (via intraperitoneal injection)[86], conditions conducive to future biomedical applications need to prove biocatalytic activity within the circulatory system (via intravenous injection). This is a significant obstacle to overcome given the more complex composition of blood, which houses hundreds of different serum proteins and metabolites. To tackle this challenge, this study aims to apply intravenous administration to treat subcutaneous xenograft tumors in mice (Fig. 6a).

As a preliminary study, substrate activity was first tested using a 5:4:1 mixture of blood/PBS/dioxane (Fig. 6b). For these conditions, various concentrations of prodrug **17** were incubated with Alb-Ru (200 µM). Following reaction quenching, an extra centrifugation step was needed to remove any solid precipitates. As anticipated, product yields and TON values generally suffered across varying concentrations (Fig. 6b, entry 1–3). However, it was postulated that this is an effect of unwanted interactions with serum components, rather than the quenching of biocatalytic activity of the Alb-Ru complexes. To prove this, a reaction was setup where Alb-Ru was pre-incubated in the blood mixture at 37 °C for 10 min. Prodrug **17** was then added, followed by yield determination after 2 h. Comparison of the activities with and without preincubation

reveal them to be similar within error (Fig. 6b, entry 3–4). Although it is expected that prolonged exposure to blood will eventually quench Alb-Ru, it is encouraging to see that the albumin scaffold does indeed have a protecting effect of the bound ruthenium catalyst, as previously observed[66,67].

Finally moving onto animal experiments, four groups of mice ($n = 5$) were arranged to receive various conditions of the planned prodrug therapy. Control groups consisted of a saline solution (vehicle), **17** only, and GArM-Ru only. With the treatment group, both **17** and GArM-Ru were sequentially administered. To ensure reactions proceed in vivo, all compounds were added via intravenous injections. Whenever necessary, a saline solution was used to replace compounds in the injection protocol. While monitoring tumor growth over a period of 20 days (Fig. 6c), a clear depreciation in the rate of tumor growth can be seen for the treatment group compared to the controls. To consolidate this data, all mice were sacrificed at day 20 and their tumors were extracted (shown in Fig. 6d). Quantification of tumor weights then helped to show the statistically reduced tumor burden of the treatment group compared to each of the three control conditions (Fig. 6e).

## Discussion
Through this study, two principal objectives were met. The first was the design and development of a synthetic prodrug that showed excellent in vivo anticancer activities in cell- and animal-based studies. Given the general difficulty in producing working examples of synthetic prodrugs, a train of thought described as retrosynthetic prodrug design was employed. In its approach, emphasis is mainly placed on first identifying useful chemical

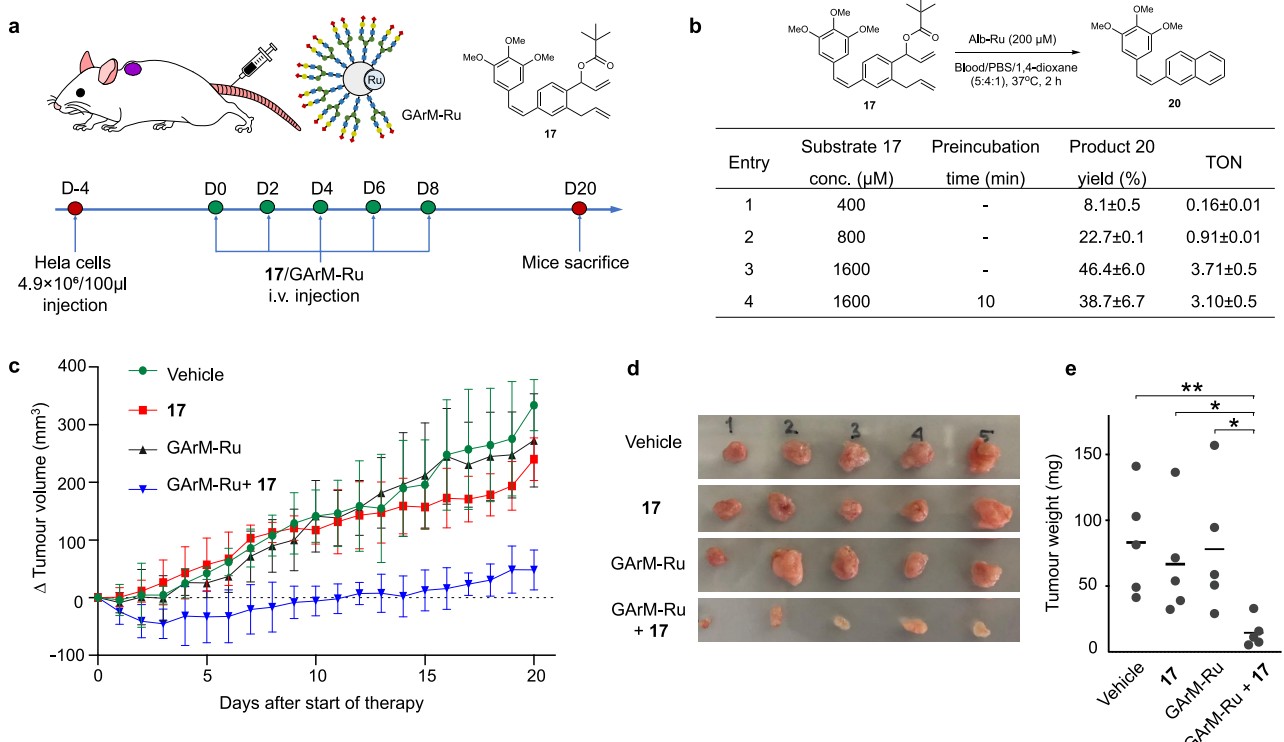

**Fig. 6 In vivo synthetic prodrug therapy against HeLa tumor growth in mice. a** To highlight the biocatalytic activity of the GArM-Ru complex, the objective was to apply prodrug therapy via intravenous administration to treat subcutaneous xenograft tumors in mice. **b** Reactivity studies of prodrug **17** with Alb-Ru in mixtures of 5:4:1 blood/PBS/dioxane. Product yield was determined by HPLC analysis. Reactions were performed in triplicate. **c** Measurements of tumor size (mm$^3$) in mice over time; Tumors were initially implanted in mice and developed over 4 days before therapy. Dosages were applied in 5 injections spread out over 8 days. **d** Extraction and visual comparison of tumor tissues shows the extent of growth inhibition 20 days after the start of therapy (n = 5). **e** Comparison of measured tumor burden (via weight) 20 days after the start of therapy (n = 5). P values were determined using a two-tailed Student's *t*-test. **\*\*P = 0.0067 (Vehicle), \*P = 0.0274 (Prodrug 17), \*P = 0.0243 (GArM-Ru); \*P < 0.05, \*\*P < 0.01, and \*\*\*P < 0.001. Data in **c** are represented as mean value ± SD, n = 5 biologically independent samples.

reactions that can lead to the formation of a drug's core backbone structure (i.e. pharmacophore). Once identified, a literature search can be conducted to find relevant bioactive molecules. Fundamentally, this is different from classical retrosynthetic analysis, which often starts with a target molecule before searching for applicable reactions. Besides the initial design, other factors also need to be considered to maximize prodrug activation. For example, this study also considered (1) factors that increase aromatization at physiological pH, (2) factors that improve substrate properties to the biocatalyst, (3) factors that decrease the intrinsic bioactivity of prodrugs, and (4) factors that improve hydrolytic stability in blood. Using this approach, prodrug **17** was developed, which is likely why it displayed such excellent in vivo activity. Despite these positives, there is clearly still room for improvement. One issue relates to the high concentration of **17** required to elicit a therapeutic effect. Although this is not a critical handicap for **17** since it is much less bioactive compared to its intended drug, future studies may need to investigate elements that can improve activation (such as to spatially design glycosylated artificial metalloenzymes to ensure substrate entry cannot be blocked).

The second objective of this study was to highlight the biocatalytic activity of the GArM-Ru complexes. Previously, our group found that in order to elicit therapeutic effects in mice, GArM complexes were best administered intraperitoneally[86]. This was largely done to avoid the complex nature of the circulatory system, which leads to greatly decreased yields compared to in vitro conditions. Although previous success with intravenously injected GArM complexes was done with tissue labeling in mice[87], this

application only required enough in vivo reactivity for detection, not to elicit a tangible biological effect. However, with the design of prodrug **17**, its mixture of good catalytic activity, good in vivo stability, and neutral bioactivity gave a concrete opportunity to test its in vivo therapeutic effects using a system of intravenously injected GArM complexes. With the promising data obtained, this study represents a leading example of the prospects that artificial metalloenzymes can be translated for usage in biological systems.

Overall, this study joins a growing list of examples in the field of in vivo metal catalyzed reactions. By showing a working example of the methodology that led to a synthetic prodrug capable of in vivo activation in mice, it is expected that interest in this field will grow and eventually spur the development of not only other cases of synthetic prodrugs, but also therapeutic applications of artificial metalloenzymes.

## Methods

**General information**. Reagents and materials were acquired from various companies (Wako Chemicals, TCI, Sigma-Aldrich, or Fisher Scientific) without further purification. Fmoc protected complex N-glycans were supplied by Glytech Inc. Human serum albumin (≥98%) was purchased from Sigma-Aldrich, and then further purified with an additional procedure[68]. All air-sensitive experiments were done under nitrogen gas. The anhydrous solvents used in this study, which were acquired from FUJIFILM Wako Pure Chemicals, include tetrahydrofuran, dichloromethane, benzene, 1,4-dioxane, acetone, dimethylformamide, and chloroform. Ultrapure water was dispensed by a Milli-Q Advantage® A10 Water Purification System from Merck Millipore.

Thin layer chromatography silica gel 60 Å F$_{254}$ glass plates from Merck Millipore were used. $^1$H and $^{13}$C NMR characterizations were done using either a JEOL AL300 (300 MHz) or AL400 (400 MHz) instrument. NMR spectroscopy was

collected using JNM-AL version 6.0 software. In terms of internal standards, identified solvent peaks include δH 0.00 for TMS, δH 7.26 and δC 77.16 for CDCl₃, δH 3.31 and δC 49.15 for CD₃OD. Low weight mass characterizations were done using high-resolution mass spectra (HRMS) done on a Bruker MicroTOF-QIII spectrometer® with electron spray ionization time-of-flight (ESI-TOF-MS). MS data was collected on Bruker Daltonics Hystar version 3.2 SR4 software. High weight mass characterizations (i.e. proteins) were done using matrix-assisted laser desorption ionization (MALDI-TOF) mass spectrometry analysis on a Shimadzu Benchtop Linear MALDI-8020 Mass Spectrometer. MALDI-TOF spectrometry data was collected on Shimadzu MALDI Solutions (Version 2.6.0). Sample preparations used sinapic acid as a matrix.

**HPLC analysis.** To identify compounds from reaction mixtures, reverse-phase HPLC was principally used with a Shimadzu system made of two LC-20AP pumps and a SPD-20AV photodiode array detector (set to 220 and 254 nm). HPLC data was collected on LabSolutions Realtime Analysis version 5.81 SP1 software. Samples were eluted using an analytical 4.6 × 250 mm Cosmosil 5C₁₈-AR-300 column from Nacalai Tesque using a combination of mobile solvents A (H₂O with or w/o 0.1% TFA) and B (acetonitrile with or w/o 0.1% TFA). The HPLC methods outlined in Supplementary Table 1 were used to identify products **2b**, **13**, **20** and **4b** (HPLC method 1), product **7b** (HPLC method 2), product **8b** (HPLC method 3) and products **5b**, **6b** and **9a** (HPLC method 4). The retention times of all compounds of interest were noted, and then identified through mass spectrometry. Product peaks were integrated and then compared with calibration curves (Supplementary Figs. 5–15) to quantify yields. Example HPLC traces of reaction mixtures, along with their corresponding product and substrate standards are shown in Supplementary Figs. 16–32.

**Preparation of compounds.** Catalyst **1**, substrates and related compounds used in this study were prepared as described in the Supplementary Methods.

**Preparation of Alb-Ru.** To prepare the artificial metalloenzyme used in this study, the coumarin-linked Grubbs catalyst **1** was anchored into the hydrophobic binding pocket of albumin. In this protocol, a solution of **1** (66.8 nmol, 167 μL from 400 μM stock solution in dioxane) in PBS buffer pH 7.4 (1336 μL) was first prepared, followed by the addition of albumin (50 nmol, 167 μL from 300 μM stock solution in PBS). The reaction mixture was then mildly mixed and incubated at 37 °C for 1 h. To remove unbound catalyst, the solution was concentrated and washed with PBS buffer using Amicon® Ultra Centrifugal Filters (30 kDa). Required concentrations of Alb-Ru were then prepared accordingly.

**Preparation of GArM-Ru.** To prepare the glycosylated artificial metalloenzyme used in this study, a solution of α(2,6)-Sia-aldehyde **21** (18.2 μmol, 4.8 mL in DMSO) in water (14.4 mL) was first prepared, followed by the addition of albumin (1200 nmol, 4.8 mL from 250 μM stock solution in water). The reaction mixture was incubated at 37 °C for 24 h. To remove unreacted aldehyde probe, the solution was concentrated and washed with water using Amicon® Ultra Centrifugal Filters (30 kDa). Confirmation of glycan linkages were made via MALDI-TOF-MS (positive mode), which detected an average molecular weight of 90 kDa. This implies that an average of 7.7 molecules of α(2,6)sialic acid terminated complex glycan were attached to the surface of each protein. In the next stage, a solution of catalyst **1** (1600 nmol, 4 mL from 400 μM stock solution in dioxane) in PBS buffer pH 7.4 (32 mL) was made. The prepared glycosylated albumin (1200 nmol, 4 mL from 300 μM stock solution in water) was then added, and the mixture was mildly mixed and incubated at 37 °C for 1 h. To remove unbound catalyst, the solution was concentrated and washed with PBS buffer using Amicon® Ultra Centrifugal Filters (30 kDa). Required concentrations of GArM-Ru were then prepared accordingly.

**Ring-closing metathesis/aromatization activity studies.** Reactions (done in triplicate) were generally carried out in Eppendorf tubes containing varying conditions of substrate (**2–12**, **14–19**) and catalyst (Alb-Ru or **1** only) in desired solvents (CDCl₃ or 1:9 dioxane/PBS buffer pH 7.4). Mixtures were typically incubated for 2 h at 37 °C, followed by quenching of the catalyst using 1-dodecanethiol. Solutions were then centrifuged for 10 min at 16,900 × g and the supernatant was analyzed by HPLC. For cases of higher substrate concentrations, 500 μL of the 1 mM 1-dodecanethiol stock was used, and 50 μL of the supernatant was injected. For cases of lower substrate concentrations, 600 μL of a 1 mM 1-dodecanethiol stock was used, and 100 μL of the supernatant was injected.

For Fig. 2, varying substrates **2–12** (5 μL of 80 mM stock in 1,4-dioxane) were mixed with catalyst **1** (5 μL of 800 μM stock in 1,4-dioxane) and PBS buffer (90 μL). For Fig. 3, substrate **12** (10 μL of 40, 20, or 10 mM stock in 1,4-dioxane) was mixed with Alb-Ru (40 μL of 25, 50, or 100 μM stock in PBS) and solvent (50 μL of PBS or D-MEM media). For Fig. 4, substrates **16–19** (20 μL of 50, 100, 500, 1000, or 2500 μM stock in 1,4-dioxane) were mixed with Alb-Ru (20 μL of 100 μM stock in PBS) and PBS solution (160 μL). For Supplementary Fig. 33, varying substrates **2–9** (10 μL of 40 mM stock in 1,4-dioxane) were mixed with Alb-Ru (40 μL of 100 μM stock in PBS) and PBS buffer (50 μL). For Supplementary Table 5, another set of conditions used substrate **16–19** (10 μL of 100, 500, 1000, 2500, or 5000 μM stock

in 1,4-dioxane) mixed with catalyst **1** (10 μL of 200 μM stock in PBS) and PBS solution (180 μL).

**Blood-based activity and stability studies.** Blood was taken directly from the tail vein of 6-week-old female nude mice BALB/cAJcl-nu/nu b, mixed with sodium citrate solution in PBS (blood citrate concentration 40 mM) and stored at 4 °C before usage. For activity-based studies, blood (100 μL) and Alb-Ru stock solution (80 μL of 500 μM stock in PBS) were first placed into an Eppendorf tube, followed by the timed addition (0 or 10 min wait) of substrate **16–18** (20 μL of 4 mM, 8 mM, or 16 mM stock in 1,4-dioxane). The mixture was then incubated for 2 h at 37 °C, followed by 1-dodecanethiol quenching (1000 μL of 1 mM stock in MeOH). Solutions were centrifuged twice for 10 min at 16900 x g and then the supernatant was analyzed by HPLC (50 μL injection). For stability-based studies, blood (100 μL) and PBS buffer (80 μL) were first placed into an Eppendorf tube, followed by the addition of either substrates **16–18** (20 μL of 4 mM stock in 1,4-dioxane). The mixture was then incubated for 2 h at 37 °C, followed by 1-dodecanethiol quenching (1000 μL of 1 mM stock in MeOH). Solutions were centrifuged twice for 10 min at 16,900 × g and then the supernatant was analyzed by HPLC (50 μL injection).

**Kinetic studies.** To carry out Michaelis–Menten kinetics, reactions were setup but then quenched and analyzed following short time interval (2 min). In an Eppendorf tube, substrates **2**, **12**, **14–18** (10 μL of 0.63, 1.25, 2.5, 5, 10, 20 mM stock in 1,4-dioxane) were mixed with Alb-Ru (10 μL of 100 μM stock in PBS) and pre-warmed PBS buffer (80 μL). Mixtures were then quickly incubated at 37 °C, followed by quenching of the catalyst using 1-dodecanethiol (500 μL of 1 mM stock in MeOH). Solutions were centrifuged for 10 min at 16,900 × g and 50 μL of supernatant was analyzed by HPLC.

**Docking studies.** To carry out molecular docking studies, the three-dimensional X-ray structure of tubulin (entry 5LYJ) was first taken from the Protein Data Bank. Calculations were then performed using Autodock 4.2.6 software[88], which is accessible via AutoDockTools (ADT version 1.5.6)[88]. Structures of small molecules were initially cleaned and saved in PDB format with Discovery Studio Visualizer (version 17.2.0.16349). In terms of protein setup, AutoDockTools set up tubulin with all hydrogens added and non-polar hydrogens merged to their respective carbon atoms. Default parameters were typically used where the exhaustiveness of the global search was set to 10. The docking space was a grid box (50 × 50 × 50) centered at the colchicine binding pocket, which was at 13.042, 9.202, −17.745 (x,y,z) with a spacing of 0.375 Å.

**Cell culture.** Cell lines were obtained from the RIKEN Cell Bank and typically incubated at 37 °C with a 5% CO₂ humidified environment. HeLa S3 (human cervical cancer cells), A549 (human adenocarcinomic alveolar basal epithelial cells), and PC-3 (human prostate cancer cells) were cultured in Dulbecco's Modified Eagle's medium (DMEM) (Wako-Fujifilm), while MCF-7 (human breast cancer cells) were cultured in RPMI 1640 medium (Wako-Fujifilm). Both culture media types were supplemented with 10% fetal bovine serum (Biowest) and 1% penicillin–streptomycin (Gibco).

**Cell viability experiments.** To quantify cell growth, the CellTiter 96 AQueous One Solution cell proliferation assay from Promega was employed. Based on cell titration experiments to ensure controls do not reach the stationary phase at the time of analysis, ~10³ cells were plated in each well of 96-well Falcon microplates. Following overnight growth and media removal, cells were then incubated with various conditions and allowed to grow for 96 h at 37 °C. For evaluation of GR₅₀ values, various concentrations of drug **20** (Supplementary Fig. 36) (0.00003, 0.00006, 0.00012, 0.00024, 0.0005, 0.001, 0.002, 0.004, 0.008, 0.016, 0.032, 0.064, 0.128, 0.256, 0.512, 1, 2, 4, 8 μM), prodrugs (Supplementary Figs 36, 37) **16**, **17** (0.008, 0.016, 0.032, 0.064, 0.128, 0.256, 0.512, 1, 2, 4, 8, 16, 32, 64 μM) and **19** (0.0005, 0.001, 0.002, 0.004, 0.008, 0.016, 0.032, 0.064, 0.128, 0.256, 0.512, 1, 2, 4, 8) were tested. For in cellulo activation studies (Fig. 5b and Supplementary Fig. 38), varying concentrations of prodrug **17** (0.128, 0.256, 0.512, 1, 2, 4, 8, 16, 32, 64 μM) was incubated with a fixed concentration of Alb-Ru (0.25 or 0.5 μM). For another experiment (Fig. 5c and Supplementary Fig. 39), varying concentrations of Alb-Ru or GArM-Ru (0, 0.03, 0.06, 0.13, 0.25, 0.5, 1, 2, 4, 8 μM) were incubated with a fixed concentration of prodrugs **17** (4 μM). To initiate viable cell detection, media was replaced with a solution of MTS reagent (15 μL) in growth media (85 μL). Following a further 2 h incubation at 37 °C, end-point absorbance was acquired at 490 nm via a SpectraMax iD3 multi-mode microplate reader (Molecular Devices), data was collected using SoftMax Pro version 7.0.3 software. In terms of controls, 100% growth was taken from cells incubated with 1% DMSO in growth medium. GR₅₀ values were calculated as described in literature[76].

**Animal experiments.** All animal experiments were carried out with approval by RIKEN's Animal Ethics Committee. In general, mice were anesthetized with 2.5% isoflurane in oxygen at a flow rate of 2.5–3.0 L/min. HeLa S3 xenograft tumors were established in 6 week-old female nude mice BALB/cAJcl-nu/nu by

subcutaneous injection of cells (approximately $4.9 \times 10^6$ cells in 100 mL of unnourished DMEM) into the right shoulder. Tumor growth was monitored while mice were housed in a facility with controlled temperature, salinity, aeration, and a standard 12 h light/12 h dark cycle. Stock samples were prepared as follows. Prodrug **17** was first dissolved in DMSO (60 μL), followed by the addition of Tween 80 (120 μL), and then a 0.9% saline solution (1020 μL). GArM-Ru was prepared in PBS buffer (180 μL), followed by the addition of a 0.9% saline solution (1020 μL). On day 4 following tumor implantation, the HeLa S3-bearing mice were randomly divided into 4 groups: vehicle control (group 1, $n = 5$); prodrug **17** only (group 2, $n = 5$); GArM-Ru only (group 3, $n = 5$); GArM-Ru and **17** (group 4, $n = 5$). For each mouse, total injection volumes never exceeded 100 μL. Following these guidelines, group 1 received saline (100 μL); group 2 received 44.1 mg/kg of **17**; group 3 received 116.4 mg/kg of GArM-Ru; and group 4 received 116.4 mg/kg of GArM-Ru, followed by 44.1 mg/kg of **17**. Treatments were done every other day for 5 total injections. The tumor volume and bodyweight of the mice were recorded every day until day 20 post-injection. Tumor volume was quantified using a caliper and calculated as $width^2 \times length \times 0.5$. On day 20 post-injection, mice were sacrificed and their tumors were excised, imaged, and weighed.

**Reporting summary**. Further information on research design is available in the Nature Research Reporting Summary linked to this article.

## Data availability

Experimental data that supports these results and other findings are available with the article, and can also be obtained from the corresponding author upon reasonable request. The source data underlying Figs. 2; 3b–d; 4b–d; 5a–c; 6b–c, e, Supplementary Figs. 4, 33, 37, 39, and Supplementary Table 5 are provided as a Source Data file. Source data are provided with this paper.

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

## Acknowledgements

We would like to thank Glytech Inc. for supplying various *N*-glycans. This work was financially supported by the AMED Grant JP15KM0908001, a research grant from the Astellas Foundation, Mizutani Foundation for Glycoscience, and JSPS KAKENHI Grant Numbers, JP21H02065 and JP21K19042 (to K.T.). This work was also supported by the Kazan Federal University Strategic Academic Leadership Program.

## Author contributions

Preparation of reagents, reactivity and kinetic studies were performed by I.N. and I.S. Modeling studies were carried out by K.V. Cell based experiments were done by I.N. and K.V. Mice experiments were carried out by P.A. The manuscript was written by I.N. and K.V., and checked by K.T. and A.K. The research was directed and supervised by K.T.

## Competing interests

The authors declare no competing interests.
