## [Peer Review File · Nature Communications]

Editorial Note: This manuscript has been previously reviewed at another journal that is not operating a transparent peer review scheme. This document only contains reviewer comments and rebuttal letters for versions considered at Nature Communications

REVIEWERS' COMMENTS

Reviewer #1 (Remarks to the Author):

The authors addressed the issues previously raised in a satisfactory manner. The flow of the manuscript with the newly included data has improved much and I believe this contribution will be highly valuable for the readership of Nature Communications. I therefore recommend publication of this manuscript and I am looking forward to the follow-up studies of this very interesting system.

As a small side note, the authors might want to cite a review that has been published during the course of the review-process (Schwaneberg et al., Nature Catalysis, 4, 814-827, 2021)).

Reviewer #2 (Remarks to the Author):

The careful and complete response to this reviewer's comments is greatly appreciated and no further issues are raised.

Reviewer #3 (Remarks to the Author):

The resubmitted version of the manuscript is much improved. The authors have taken great care in addressing all queries. The restructured manuscript has a more logical order and the text is easier to follow. In addition, the relevance of the findings is explained more clearly. Good use of revised Figures is being made. The manuscript is now suitable for publication.

On behalf of my co-authors and myself, we would like to thank the reviewers for peer-reviewed of our manuscript.

In responded to reviewer 1 recommendation, we added reference 66 (page 3, line 91).

----- **Reviewer 1** -----

1 COMMENT: As a small side note, the authors might want to cite a review that has been published during the course of the review-process (Schwaneberg et al., Nature Catalysis, 4, 814-827, 2021)).

RESPONSE: We would like to thank the reviewer for this suggestion. As advised, we have added a reference (page 3, line 91).